# Socio-Economic Determinants of Dental Service Expenditure: Findings from a French National Survey

**DOI:** 10.3390/ijerph19031310

**Published:** 2022-01-25

**Authors:** Anne-Charlotte Bas, Sylvie Azogui-Lévy

**Affiliations:** 1Dental Public Health Department, Faculty of Dentistry, Paris University, 75006 Paris, France; sylvie.azogui-levy@u-paris.fr; 2Inserm U1018, Centre de Recherche en Épidémiologie et Santé des Populations, 94807 Villejuif, France; 3Educations and Health Practices Laboratory (LEPS EA 3412), Faculty of Medicine, Paris 13 University, 93017 Bobigny, France

**Keywords:** oral health, healthcare disparities, access to care, health expenditure, health service research, health economics

## Abstract

(1) Background: This study investigated how individual enabling resources influence (i) their probability of using dental services and (ii) consumers’ expenditure on dental treatment. (2) Methods: Data were derived from a self-administered national health survey questionnaire and from expenditure data from national health insurance. Multiple linear regression methods were used to analyze entry into the dental health system (yes/no) and, independently, the individual expenditure of dental care users. (3) Results: People with the highest incomes were more likely to use dental service (aOR = 1.59; 95% CI = 1.28, 1.97), as were those with complementary health insurance and the lowest deprivation scores. For people using dental services, good dental health status was associated with less expenditure (−70.81 EUR; 95% CI = −116.53, −25.08). For dental service users, the highest deprivation score was associated with EUR +43.61 dental expenditure (95% CI = −0.15; 87.39). (4) Conclusion: Socioeconomic determinants that were especially important for entry into the dental health service system were relatively insignificant for ongoing service utilization. These results are consistent with our hypothesis of a dental care utilization process in two steps. Public policies in countries with private fees for dentistry should improve the clarity of dental fees and insurance payments.

## 1. Introduction

Especially marked horizontal social inequalities exist in access to dental care: individuals with the same oral health status have different degrees of access to care according to their social status. Dental care needs are unmet for financial reasons for 18% of the French population [1], 15% in Europe [2] and from 16 to 35% depending on dental insurance coverage in the U.S. [3]. Although much has been written on the subject of financial barriers to health-care access, there is little research on dental expenses and the mechanisms by which these barriers hinder utilization.

Economic studies show that income, complementary health insurance and socioeconomic status (including educational level, income category and deprivation) all affect access to dental care [4,5,6,7,8,9,10]. Dental pain, fear of dental care, oral health knowledge and dental service trajectories have also been related to dental service utilization [4,7,10]. Perceptions of need for dental treatment have been associated with a low socioeconomic status [4,5,6,11] and non-utilization of dental services [4,6].

Most dentists in France are in private practices. Patients are free to choose their own dentist. The patient usually pays the entire fee and is subsequently reimbursed by the National Health Insurance Fund (NHI). The NHI establishes a fixed, mandatory price for standard dental care such as fillings, descaling or endodontic treatment. The patient is reimbursed 70%, and the rest is covered by complementary insurance policies. For prosthetic treatments, the NHI establishes a basic price that is much lower than the cost price. The dentist is allowed to charge a higher price than that authorized by the NHI, while the complementary insurance covers an amount that varies according to the type of contract but not always the full amount. Private complementary health insurance policies offer different levels of reimbursement. Concerns about social equality led to the development of a public complementary health insurance program for people living in France with very low incomes (i.e., annual income under EUR 8951). This health insurance has a set of fully regulated fees, and its beneficiaries have no payments to make. Periodontal and implant treatments or maintenance care engender private fees and are excluded from the NHI scheme, except for people with facial damage or specific rare diseases. In general, there are very few chronic illnesses that generate complementary dental payments from the NHI. Despite the well-known relationship between certain chronic illnesses and oral diseases, there was no association between dental and medical expenses over a 4-year period in the French population with chronic diseases [12]. There is a lack of recent and reliable oral health data for the French population, but almost 13% of the population studied in this paper reported more than four missing teeth.

The literature on dental care utilization applies methods originating from epidemiology and public health research. An example of this type of research is Andersen’s behavioral model of health care utilization (recently updated by Andersen in 2014 [13]). As defined by Lurie and then Lombrail [14,15], care utilization is a two-step process that depends on the success of both primary and secondary access. Primary access (i) is the first visit to or entry into the dental health system, while secondary access (ii) is ongoing treatment.

The aim of this study was to investigate the financial barriers that most strongly affect dental health service utilization in a partially state-regulated health system. The research hypothesis was that determinants differ for (i) the probability of using dental services and (ii) the consumer’s expenditure on dental treatment.

## 2. Materials and Methods

### 2.1. Data Sources

This study combined data from two sources: (j) the 2010 cross-sectional Health and Health Insurance Survey (ESPS, conducted by Institute for Research and Information in Health Economics (IRDES)) and (jj) 2010 administrative health consumption data from the NHI. The ESPS survey includes data about health insurance, health status and socioeconomic status, and it stopped in 2012. The study variables were last available in 2010, which explains the age of the files. We decided to accept this because the association of health status and socioeconomic characteristics with administrative data on oral health care consumption is particularly rich and very rare and enables the exploration of mechanisms that are still relevant. Furthermore, this data linkage is quite remarkable because of the simultaneity of social, health and well-established expenditure data. Data were linked using the individual NHI number in the public organizations responsible for this treatment. Post-survey data matching was successful for 49% of the ESPS sample. Weights provided by IRDES were applied on the basis of the data linkage in order to remain representative of the adult French population. The total sample comprised 6222 individuals. The choice of explicative variables was based on determinants from the last systematic review [10].

### 2.2. Outcome Measures

The analysis comprised two phases: step 1—the probability of accessing dental services and (step 2) consumers’ expenditure on dental treatment.

Utilization of dental services was a dummy variable (yes/no), according to whether the cohort members used any dental service in 2010. This variable relates to the entry into the dental care system in the two-step utilization process. The second outcome variable studied was the amount of total individual dental expenditure for cohort members with at least one dental visit during the year 2010. Every dental procedure was counted, regardless of the treatment. This variable provides the second step of the process: ongoing dental treatment after entry into the dental care system.

### 2.3. Covariates

The predisposing characteristics were gender and age in six categories. The need factor was oral health status. We created a composite “oral health status” dummy variable (good/poor). This composite variable was based on both a subjective indicator, individual perceived dental health status, and the self-reported number of non-wisdom, permanent teeth lost, which was the most objective dental health indicator available. Cohort members were considered to have good dental health status if they reported good or very good dental health and fewer than 5 missing teeth. The others were classified as having poor dental health status. This composite variable has previously been successfully used in a French study on dental care access [16]. A geographic variable described the type of residential area, urban or rural, in 5 categories, according to the population numbers in the area.

Three socioeconomic determinants of utilization concerned enabling resources: the type of complementary health insurance (public, private or none), household income (in 5 categories according to French population income quintiles; cat.1 is the lowest category) and EPICES deprivation score. These were considered as probably related to the ability to pay for health services. The EPICES deprivation score was used to assess social support [17]. The EPICES score is an assessment of precariousness and health inequalities. It is an individual indicator of deprivation that takes the multidimensional character of deprivation into account. This score is computed from 11 items. It takes into account several dimensions of deprivation: employment, income, educational level, occupational category, housing, family composition, social ties, financial difficulties, life events and perceived health. It includes questions about social support: “In case of difficulties (financial, family, health.), are there people among your family or friends on whom you can count to provide you with material assistance (including a loan)?”. The sum of the weighted 11 responses yields the EPICES score, continuously from 0 (no deprivation) to 100 (maximum deprivation). This variable was introduced as follows:Cat I = [0–7.10]: well-off;Cat. II = [7.10–16.56]: fairly well-off;Cat III = [16.56–30.17]: not deprived but just under the deprivation threshold;Cat IV = [30.17–48.52]: deprived;Cat V = [48.52–100]: very deprived.

This index made it possible to identify socially and/or medically vulnerable populations that are not detected by the socio-administrative criteria [17]. It also provides better information about economic matters than the Duke Social Support Index [18].

At individual level, we tested and rejected other explanatory variables, including general health status, numbers of persons or family members in the household, occupational category and educational level. None of these variables affected the outcomes, modified the effect of the other explanatory variables or significantly changed the pseudo-R-squared values of the models.

### 2.4. Data Analysis

Statistical analyses were conducted with Stata 15 (StataCorp). Descriptive statistics were used to explore the distribution of participants’ demographic and socioeconomic characteristics according to dental service utilization (Table 1). Because 3653 people in the sample did not consult a dentist and thus did not have any dental expenditure, the preliminary analysis used a Heckman selection model. Running this model produced the inverse Mills ratio, which indicated a very low probability of selection (*p* = 0.34) and showed that the probability of dental service utilization was independent from the differences in dental expenditure levels. This preliminary analysis led us to renounce the Heckman selection model and run the statistical analysis in two independent steps. This method is more legible and parsimonious and reduces multicollinearity. It is consistent with the theoretical framework and enabled us to study sequentially primary access for entry into the dental health system and secondary access and the amount of dental expenditure by users.

The first analysis was a logistic regression model that explains dental service utilization (Step 1). The results are presented as adjusted odds ratios (aOR) (Table 2). Multiple linear regression methods were used to examine dental expenditure and its association with covariates in the user sample (step 2). The dependent variable was the logarithm of user dental care expenditure because dental care expenditure was not normally distributed. The Z tests performed validated this choice. The results are presented as adjusted coefficients and marginal effects for easier reading (Table 2). Income and health insurance are both considered in the EPICES score. To examine the separate impact of each enabling factor, we performed multiple linear regressions for step 1 and then step 2 with one enabling factor at a time (Table 3).

## 3. Results

### 3.1. Descriptive Statistics

A total of 59% of the sample did not use dental services at all. Those who did spent an average of EUR 372 on dental care over the year (SD = 763). Individual total expenditure ranged from EUR 8 to EUR 17,970. The very high standard deviation implies the need to moderate expectations for meaningful results.

Almost 40% of the main sample could be considered as socially deprived, but this figure fell to 34% for the dental services user sample (Table 1). The same trend was observed for individuals in the user sample with complementary coverage.

### 3.2. Primary Access or Entry into the Dental Health System

Men were less likely than women to use dental health services (aOR = 0.73, 95% CI = 0.65, 0.82). The youngest age category had the lowest probability of dental care utilization. Dental care utilization was positively associated with self-reported good dental health (aOR = 1.55, 95% CI = 1.23, 1.95) (Table 2).

The higher the social deprivation score, the lower the probability of using dental services (maximum deprivation score class: aOR = 0.52, 95% CI = 0.34, 0.79). People with high incomes had a notably higher probability of seeking dental care than those with the lowest incomes (aOR highest cat. = 1.59, 95% CI = 1.28, 1.97). Public complementary health insurance improved access to dental care (aOR = 2.42, 95% CI = 1.48, 3.95), and private supplemental complementary health insurance was linked to a positive trend toward significantly better access (aOR = 1.64, 95% CI = 1.02, 2.65).

### 3.3. Secondary Access or Ongoing Dental Treatment

Men had greater mean expenditure than women (*p* < 0.1), and the youngest cohort members had the lowest expenditure (*p* < 0.1). Reporting good dental health was also associated with EUR 71 less in expenditure (95% CI = −116.53, −25.08).

The highest level of social deprivation was associated with EUR +44 dental expenditure (95% CI = −0.15, 87.39). People who consulted dentists in rural areas had higher dental expenditure than those in more urban areas. 

The effects of income on the level of expenditure were neither significant nor stable. The EPICES score and health insurance were both associated with the level of dental expenditure, but social determinants interacted with each other. High social deprivation scores had the most stable and significant effect on expenditure (Table 3). It seems that enabling resources had a greater impact on step 1 than on step 2, which could be related to selection at the entrance into the care system (leading to a marked selection effect in our sample).

## 4. Discussion

In an analysis of a large representative dataset of insured individuals, socioeconomic determinants proved to have a much stronger effect at entry into the dental health system than thereafter. This contribution is doubly interesting: (i) we have identified enabling factors that enhance primary access and then limit the amount of dental care expenditure, and (ii) we have shown that the factors enabling primary access were less substantially associated with dental care expenditure. These two regression analyses can be compared only in terms of their direction and the significance of the effects. The results are consistent with our hypothesis of dental service utilization as a two-step process. The public complementary insurance scheme significantly improved access to dental care. This exploration shows the efficacy of this public health policy.

As the previous literature has shown, oral health status is the major determinant in the use of dental services [10,19]. Reporting good dental health was positively associated with initial utilization and negatively related to ensuing total expenditure. This association probably results from a specific dental visit trajectory pattern associated with dental health status. Check-ups are less expensive than prosthetic treatments. As a result, users seeking to maintain good dental health via regular check-ups are more likely to have lower expenditure than occasional users. According to Grembowski et al. [20] and Worsley et al. [5], people with good dental health go to the dentist for check-ups, while people with poor dental health status tend to see dentists to deal with the worst problems, and this second category could benefit from more information about dental service’s usefulness. The available data comprised little information on different oral health variables, so it was not possible to analyze expenditure in relation to specific types of dental care, differentiating surgery, prevention, conservative care, and so on. Nonetheless, as previous authors have reported, these results suggest a positive association between the use of preventive care and good oral health and thus encourage the development of incentives to seek preventive dental care [21,22]. This health-care behavior avoids the deterioration of dental health and the need for expensive dental care [23,24]. This study takes very few enabling factors into consideration, but the literature shows that poor knowledge about oral health and fear of dental costs leads to underutilization [1,7,10]. Non-use at step 1 could also be related to oral health literacy: misunderstanding of the health care system, pricing, the need for regular dental care, etc. [25,26] Improving oral health literacy improves patients’ empowerment and their ability to seek care. Policy makers could increase information about dental service’s usefulness and dental cover in order to have an impact on health oral care habits that amplify oral health inequalities. The literature on the association between access to oral and medical health services is sparse, and dental expenditure is very different from medical expenditure in that it is not correlated with general health status or chronic disease medical care, as NHI participates more in medical costs, and dental care is often utilized at one point in time [12].

The analysis focuses on enabling resources, and entry into the system appears to be the key moment at which financial barriers act on dental care utilization. This finding is in line with previous studies that describe a “pro-rich” distribution of dental service use, a positive impact of complementary health insurance on dental care utilization [10] and a positive association between dental fees and unmet dental care needs [1]. The results allow us to go further: socioeconomic determinants stop people at the entry to the dental health system, but those who succeed in entering then leave these determinants on the doorstep. Policy makers should consider this specific financial barrier at entry into the oral health system, given that public complementary health insurance is efficient at both stages in access. The socioeconomic variables we studied play a lesser role in determining the level of dental expenditure. Income seems to be less important than other socioeconomic variables. It seems that people take their financial capacities into account when deciding to see a dentist, but the dentist appears to decide on the treatment plan and the ensuing costs, which are uninfluenced by the patient’s socioeconomic characteristics.

The study has several limitations. The data were collected a decade ago. Given the absence of change in the regulations in force, we think these findings are still relevant today. The cross-sectional nature of the study means that no causal relationships can be inferred. We weighted the data to deal with sample depletion due to matching and the different response rates. These missing data could have resulted in bias, especially as it is likely that the poorest households disproportionately more frequently failed to answer questions [27]. The association between utilization of dental care and social characteristics could thus have been underestimated. Nonetheless, the ESPS data provide a large nationally representative sample, which is quite rare in the relevant literature. Our major concern in this analysis of dental health expenditure is the vast group of non-users. This group is clearly not homogeneous, but the model tells us nothing about the motivations of non-users. Some do not use dental services because they have no need for them, while others have needs but are merely unable to meet the costs. Unfortunately, our need-for-dental-care indicator is not precise enough to allow us to conduct more specific statistical analyses. There is a need for reliable and recent data on oral health status in France. The lack of oral health data concerns every field of oral health research. Consequently, there is no reliable assessment of oral health in the French population to support public health policies.

## 5. Conclusions

In this study, socioeconomic deprivation appears to be a greater barrier to seeking care than it is to pursuing ongoing treatment. Public complementary health insurance was a significant factor in improving access to dental care. What improvements are possible given that entry into the system is the key moment when financial barriers hinder access to dental care? Both income and social deprivation could be related to knowledge about oral health and the health system. When dentists charge more than the standard fee for their services, as is possible in France, fear of the costs can lead patients to refuse to pursue care, as can the fear of painful dental procedures. Campaigns for more clarity about dental fees could facilitate access to dental services. It is especially important to inform the French population about the numerous existing “no out-of-pocket” options, for these are little known and underused [28].

## Figures and Tables

**Table 1 ijerph-19-01310-t001:** Descriptive statistics of the explanatory variables for the main sample (N = 6222) and the service user sample (N = 2569).

	Main Sample	User Sample		Main Sample	User Sample
	N (%)	N (%)		N (%)	N (%)
Predisposing and Need Characteristics			Enabling Resources		
Gender			Income by Consumption Unit ^2^		
Men	2912 (46.80%)	1111 (43.25%)	Lowest category	757 (21.06%)	294 (18.89%)
Women	3310 (53.20%)	1458 (56.75%)	2nd category	674 (18.76%)	310 (19.94%)
Response rate	6222 (100%)	2569 (100%)	3rd category	682 (18.99%)	323 (20.76%)
Age in years			4th quintile	720 (20.03%)	4th category
16–25	677 (10.88%)	236 (9.19%)	Highest category	760 (21.16%)	328 (21.06%)
26–36	968 (15.5%)	388 (15.10%)	Response rate	3593 (57.75%)	1555 (60.53%)
37–45	1200 (19.29%)	547 (21.29%)	Complementary health insurance		
46–55	1115 (17.92%)	473 (18.41%)	None	311 (5.09%)	282 (3.46%)
56–65	973 (15.64%)	456 (17.75%)	Public	612 (10.01%)	87 (9.34%)
66–100	1316 (21.15%)	469 (18.25%)	Private	5099 (83.29%)	2152 (85.36%)
Response rate	6222 (100%)	2569 (100%)	Response rate	6122 (98.39%)	2521 (98.13%)
Dental health status			EPICES score for social deprivation ^1^		
Poor	863 (19.72%)	283 (16.32%)	I—Well-off	803 (15.15%)	416 (19.07%)
Good	3514 (80.28%)	1451 (83.68%)	II—Fairly well-off	1093 (20.62%)	514 (23.57%)
Response rate	4377 (70.34%)	1734 (67.49%)	III—At the deprivation threshold	1304 (24.60%)	523 (23.98%)
Type of urban area (number of people per unit)					
Rural area	1838 (29.54%)	724 (28.18%)	IV—Deprived	1339 (25.26%)	475 (21.78%)
<20,000	1244 (19.99%)	513 (19.97%)	V—Very deprived	762 (14.37%)	253 (11.60%)
20 × 10^3^; <200 × 10^3^	1202 (19.32%)	507 (19.74%)			
200 × 10^3^; <2 × 10^6^	1230 (19.77%)	520 (20.24%)	Response rate	5301 (85.20%)	2181 (84.90%)
Paris and suburbs	708 (11.38%)	305 (11.87%)			
Response rate	6222 (100%)	2569 (100%)			

^1^ The EPICES score is a multidimensional score for social deprivation designed by the NHI and health centers. ^2^ Income by consumption units is the OECD reference measure for household income

**Table 2 ijerph-19-01310-t002:** Results for logistic regression analysis of primary access and linear regression analysis for secondary access (with marginal effects).

	**Step 1:** **Use of Dental Services** **Yes/No**	**Step 2:** **Users’ Dental Expenditure** **Log(Expenditure ≠ 0)**	
	aOR [95CI]	Coeff [95CI]	Marginal effect [95CI]
Gender (Ref. = Women)			
Men	0.73 *** [0.65; 0.82]	0.13 * [−0.01; 0.27]	−15.25 * [−2.09; 32.60]
Age in years (Ref. = 16–25)			
26–36	1.37 ** [1.03; 1.81]	0.26 * [−0.02; 0.54]	13.96 [−10.73; 38.66]
37–45	1.70 *** [1.28; 2.25]	0.54 *** [0.24; 0.83]	43.53 ** [15.26; 71.81]
46–55	1.55 *** [1.12; 2.15]	0.55 *** [0.28; 0.82]	50.69 ** [21.28; 80.10]
56–65	2.11 *** [1.49; 3.00]	0.57 *** [0.27; 0.88]	58.36 *** [26.26; 90.47]
66–100	1.25 [0.90; 1.73]	0.65 *** [0.39; 0.90]	65.91 *** [32.81; 98.99]
Dental health status (Ref. = Poor)			
Good	1.55 *** [1.23; 1.95]	−0.57 *** [−0.81; −0.32]	−70.81 ** [−116.53; −25.08]
Type of urban area (number of people per unit) (Ref. = rural area)			
<20,000	1.16 [0.95; 1.41]	−0.00 [−0.22; 0.21]	2.21 [−24.82; 29.23]
20,000; <200,000	1.30 ** [1.04; 1.62]	−0.29 ** [−0.53; −0.04]	−26.86 ** [−50.90; −2.81]
200,000; <2 million	1.21 * [0.98; 1.49]	−0.26 *** [−0.42; −0.09]	−25.67 ** [−49.56; −1.79]
Paris and suburbs	1.21 *** [1.05; 1.40]	−0.08 [−0.24; 0.07]	−6.68 [−37.06; 23.69]
EPICES score for social deprivation ^1^ (Ref. = I—Well-off)			
II—Fairly well-off	0.81 ** [0.68; 0.98]	0.22 * [0.14; 0.58]	23.63 * [0.05; 47.21]
III—At the deprivation threshold	0.79 ** [0.63; 0.99]	0.26 * [−0.03; 0.55]	25.55 * [1.42; 49.68]
IV—Deprived	0.56 *** [0.41; 0.75]	−0.01 [−0.27; 0.25]	0.62 [−23.22; 24.47]
V—Very deprived	0.52 *** [0.34; 0.79]	0.32 ** [0.05; 0.59]	43.61 * [−0.15; 87.39]
Income by consumption unit ^2^ (Ref. = Lowest category)			
2nd category	1.36 ** [1.04; 1.77]	−0.08 [−0.39; 0.23]	−10.08 [−43.64; 23.48]
3rd category	1.63 *** [1.21; 2.19]	0.04 [−0.28; 0.37]	5.707 [−29.66; 41.07]
4th category	1.59 ** [1.11; 2.27]	−0.08 [0.37; 0.21]	−9.66 [−43.58; 24.25]
Highest category	1.59 *** [1.28; 1.97]	0.06 [−0.40; 0.53]	8.16 [−18.35; 44.68]
Complementary health insurance (Ref. = No complementary health insurance)			
Public (out of pocket = 0)	2.42 *** [1.48; 3.95]	0.37 * [−0.07; 0.81]	59.83 * [4.31; 115.36]
Private	1.64 ** [1.02; 2.65]	0.10 [−0.30; 0.49]	30.88 [−10.53; 72.30]
	N = 3402 Adjusted R^2^ = 0.04 Constant = 0.17	N = 1342 Adjusted R^2^ = 0.06 Constant = 4.62	

^1^ The EPICES score is a multidimensional score for social deprivation designed by the NHI and health centers. ^2^ Income by consumption units is the OECD reference measure for household income. * Statistically significant at the 1% level. ** Statistically significant at the 5% level. *** Statistically significant at the 10% level.

**Table 3 ijerph-19-01310-t003:** Test for adjusted associations between each step of access to dental services and the enabling resources.

1. Logit Models: Use of Dental Services (Step 1)		aOR ^1^ [95CI]	
Model A1: EPICES Score ^2^	I—Well-off	Reference	N = 3402 Adjusted R^2^ = 0.03 Constant = 0.51
	II—Fairly well-off	0.79 ** [0.65; 0.96]	
	III—At the deprivation threshold	0.70 *** [0.56; 0.86]	
	IV—Deprived	0.51 *** [0.40; 0.65]	
	V—Very deprived	0.43 *** [0.33; 0.56]	
Model B1: Income ^3^	1st and lowest quintile	Reference	N = 3402 Adjusted R^2^ = 0.03 Constant = 0.21
	2nd quintile	1.44 *** [1.13; 1.82]	
	3rd quintile	1.66 *** [1.28; 2.13]	
	4th quintile	1.81 *** [1.35; 2.43]	
	Highest quintile	1.90 *** [1.55; 2.29]	
Model C1: Complementary health coverage	No complementary insurance	Reference	N = 3402 Adjusted R^2^ = 0.02 Constant = 0.15
	Public [out of pocket = 0]	1.73 ** [1.08; 2.76]	
	Private	2.19 *** [1.43; 3.37]	
**2. Linear models: dental expenditure (Step 2)**		**Coeff. ^1^ [95CI]**	
Model A2: EPICES Score ^2^	I—Well-off	Reference	
	II—Fairly well-off	0.20 [−0.14; 0.55]	N = 1342 Adjusted R^2^ = 0.05 Constant = 4.68
	III—At the deprivation threshold	0.22 [−0.05; 0.50]	
	IV—Deprived	0.03 [−0.22; 0.27]	
	V—Very deprived	0.40 *** [0.16; 0.65]	
Model B2: Income ^3^	Ref. = Lowest category	Reference	N = 1342 Adjusted R^2^ = 0.04 Constant = 5.17
	2nd category	−0.11 [−0.38; 0.15]	
	3rd category	−0.02 [−0.29; 0.26]	
	4th category	−0.18 [−0.44; 0.08]	
	Highest category	−0.06 [−0.44; 0.60]	
Model C2: Complementary health coverage	No complementary insurance	Reference	N = 1342 Adjusted R^2^ = 0.04 Constant = 4.65
	Public [out of pocket = 0]	0.62 ** [0.24; 0.99]	
	Private	0.37 * [−0.00; 0.75]	

^1^ Control variables are: gender, age, dental health status, type of urban area. ^2^ The EPICES score is a multidimensional score for social deprivation designed by the NHI and health centers. ^3^ Income by consumption unit is the OECD reference measure for household income. * Statistically significant at the 1% level. ** Statistically significant at the 5% level. *** Statistically significant at the 10% level.

## Data Availability

ESPS 2008 dataset is available for researchers by agreement with the IRDES agency.

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
