# Peer review of "Socio-Economic Determinants of Dental Service Expenditure: Findings from a French National Survey"

_ijerph, 2022, doi:10.3390/ijerph19031310_

Round 1

Reviewer 1 Report

Originality/Novelty

  1. The authors present the important topic of social determinants of dental service expenditure, but from a French context. The article touches upon important issues: oral health; healthcare disparities; access to care; health expenditures; and health services research. This paper is therefore, both original and novel. Perhaps the words: health economics, can be added to the list of Key Words, as the article does also touch upon this subject matter.

Significance of content

  1. Although focused on a French context, the article presents an example that is likely to be of relevance to the international readership. Well done to the researchers for their efforts in conducting this study, which appears to be quite complex and intricate. The article’s content is of significance, and is likely to be of more significance, given that the current global pandemic is likely to have a huge effect on oral health; healthcare disparities; access to care; health expenditures; health economics, and health services research.

Quality of presentation

  1. Overall, the paper is well-written and fairly easy to read. It is well structured and laid out. The figures, tables and statistics are satisfactory. The utilised references are fine. However, please kindly note, the language, grammar, punctuation, spelling and sentence structures within the current paper, all must be thoroughly assessed and polished to ensure a succinct and coherent read. Please obtain the necessary scientific English language reading and editing assistance if need be, so that the paper has the potential to be read enjoyably by the international readership.

  1. Some examples of discrepancies that need to be corrected:
  • Line 73-75 – “Literature on dental care utilization applies methods borrowed directly from epidemiology and public health, specifically, Andersen's behavioral model of health care utilization (most recently updated by Andersen in 2014).” This is a very longwinded sentence. Perhaps it could be changed to: “Literature on dental care utilization applied methods originate from the works of epidemiology and public health. An example of such work is Andersen's behavioural model of health care utilization (most recently updated by Andersen in 2014).”
  • Line 100 – “Other were classified as in poor dental health status.” Do you mean: “Others were classified as poor dental health status” ?
  • Line 115- “are there people among your family or friends on whom you can count to provide you….” Should this be: “are there people among your family or friends on whom you can count on to provide you….” ?
  • Line 117-118- “This variable was introduced as follow:” Should this be: “This variable was introduced as follows:” ?
  • Line 156- “..on both step of access..” Should this be: “..on both steps of access..” ?
  • Line 157- “..for step 1 then step 2..” Should this be: “..for step 1, then step 2..” ?
  • Line 207- “Analyzing a large representative dataset of insured people, we found that..” Might: “After analyzing a large representative dataset of insured people, we found that..” sound better?
  • Line 219- The sentence: “People faced increasing oral health inequalities” feels out of place. This sentence needs to be better incorporated within the surrounding text.
  • Lines 219-221- The sentence: “Because checkups are less expensive than prosthetic treatments, users engaged in maintaining good dental health then have lower expenses than opportunist users” needs to be made clearer. Perhaps you might consider the following: “Check-ups are less expensive than prosthetic treatments. So, users engaged in maintaining good dental health via regular check-ups are more likely to have lower expenses than opportunist users; the latter may require more comprehensive dental treatment to include prosthetic treatments.”
  • Line 226-227- “..our results suggest a positive association between the use of preventive care and good oral health and thus encourage the development of..” Please consider the use of a comma: “..our results suggest a positive association between the use of preventive care and good oral health, and thus encourage the development of..”
  • Line 252- “and then… there are the others.” Are the three full stops really needed?

Scientific soundness

  1. There are no obvious issues pertaining to scientific soundness. This article appears to have a satisfactory scientific basis.

Interest to readers

  1. This article shall be of interest to dentists, dental health professionals, healthcare practitioners, economists, public health workers, and policymakers, as well as others.

Overall merit

  1. Many thanks for the chance to review this manuscript. It was interesting and thought provoking to read. It does have merit. However, if the aforementioned alterations are made, then this paper will have much more strength for publication.

Author Response

Dear Reviewer,

Thank you so much for your valuable comments.

We applied all your advice in the new document.

We take particular care of these points :

  • we ask for a professional English language check;
  • Indeed, we had a contract with NHI and IRDES' institution that made sure we exploit the data in respect with ethical consideration;
  • We developed explanations about EPICES score;
  • We made more obvious that the very selective enabling factors implied a very cautious approach of the results;
  • We insist on how the results can be useful for the health policymakers.

We hope this new document is going to be suitable for publication.

Kind Regards.

Reviewer 2 Report

Very interesting research and well presented.

I would like the authors to consider a spell and grammar check and to remove the first tense from the manuscript e.g. Line 54- We hypothesize, Line 79- We propose, Line 241- Our study,.. e.t.c

Also, there are some sentences that are not making sense and the authors could rephrase them, e..g Line 36-36:"The associations linking socioeconomic status, oral health care, and oral health status 36 are complex", Line 37-38" Economics studies show that income, supplemental health insurance, and socioeconomic status all affect access to dental care in most systems, (which systems the author mean?)

Line 251-Some do not use dental services because 251 they have no need for them, and then… there are the others. 

Finally, the authors should check several pantuaction mistakes throughout the manuscript (additional puntuaction marks e.t.c)

Author Response

Dear Reviewer,

Thank you so much for your valuable comments.

We applied all your advice in the new document.

We take particular care of these points :

  • we ask for a professional English language check;
  • we developed some characteristics about the French dental health system;
  • we can't make an analysis in relation to medical expenditure because we saw in an earlier study that medical and dental expenditure are not correlated and follow a very different pattern. Nonetheless, we made this point in both introduction and discussion sections;
  • There is a need for reliable and recent data on oral health status in France. The lack of oral health data concerns every field of oral health research. Consequently, there is no reliable assessment of oral health in the French population to support public health policies. Nonetheless, we gave in the introduction the oral health assessment coming from the study's dataset.

We hope this new document is going to be suitable for publication.

Kind Regards.

Reviewer 3 Report

" Social determinants of dental service expenditure: findings from a French national survey?"

It is very interesting to investigate the financial barriers that most strongly affect dental health service utilization in a partially State-regulated health system. However, there are a few corrections that are essential to meet the standard for publication. Please refer to the following comments.

1) The authors conducted this study using insurance consumption data in French dental care. You should explain in more detail about the French dental system. Insurance consumption of medical expenses varies greatly from country to country. Is there any assistance in paying for implant treatment? Are individuals paying for medical care for maintenance after implant prosthesis? Is there any special support for payment for patients with special circumstances (such as patients who have difficulty paying after surgery for oral cancer or in poverty)? Is the payment amount fixed depending on the age?

Please elaborate and add a description.

2) Is there a correlation with the payment of medical expenses in the medical department?

If possible, add it to the results, and if analysis is difficult, add it in the discussion section using citations.

3) What is the condition of the remaining and restored teeth of modern French people? Please consider if you have any data or past reports such as the number of remaining teeth and prosthetic restorations.

Author Response

Dear Reviewer,

Thank you so much for your valuable comments.

We applied all your advice in the new document and we ask for a professional English language check.

We hope this new document is going to be suitable for publication.

Kind Regards.

Reviewer 4 Report

The authors present an analysis of combining two (French) data bases to explore the associations of dental service usage as a two-stage process; of firstly, entering into the dental health system (the primary step), and secondly, maintaining an on-going relationship with the dental health system (the secondary step).  The first dependent variable - utilization of dental services - was derived from positive responses to whether the participant in the 2010 French National Health and Health Insurance Survey (ESPS) had "used any dental service in 2010".  The second dependent variable - total individual dental expenditure - was the accumulated costs of "every dental procedure... regardless of the treatment" for each participant who had used a dental service in that year.  Fifty-nine (59) per cent of the ESPS respondents did not report using a dental service during 2010. A total "user of dental services" sample on 2,569 participants formed the combined data base for the analyses.

The key explanatory variables (other than age, and gender) were largely derived from previous work by the authors and a socio-economic estimate of income quintiles - the EPICES deprivation score - a five category ranking of social and income attributes of an individual.

I am not in a position to provide a definitive comment on the authors use and rejection of certain statistical procedures, but on face value they appear satisfactory. Although the authors indicate that informed consent was obtained from all subjects involved in the study - there was no indication that the authors themselves had received ethical approval from an appropriate institution to access and report on their findings.

The findings of those who used "dental services" showed a very high range of costs - from €8 to €17,970 - in the year of 2010. Although men were less likely to use dental services, they showed are greater mean expenditure when they did so, than women. The higher socio-economic score, the lower the probability of participants using a dental service during. However, both "public supplemental health insurance" and "private supplemental health insurance" showed a positive association with use of dental services.

Both deprivation score and income were associated with use of dental services and dental expenditure.  

While the authors are proposing an interesting approach to better understanding the relationship between socio-economic factors and the use of dental services - a two stage, entry into the dental health system, and on-going maintenance within a dental health system, plus the use of data derived from national French population studies, there are a number of weakness in the study.  The authors recognize a number of these weaknesses in their discussion but do not demonstrate the need for a very cautious approach to their conclusions, nor articulate where further research may assist public health policy architects to address the problems identified.  In fact, what appeared to this referee as an important finding was that the French government "Public supplemental health insurance" scheme significantly improved access to dental care"  (ORa =2.42, 95% CI = 1.48, 3.95).

The limitations of the measurement tools for dental health service metrics within the survey are again a major limitation.  Having to derive "oral health status" from two very subjective participant responses "perceived dental health status" and "self-reported number of non-wisdom permanent teeth" is a very weak variable to use in analyses.  The Discussion especially should reflect the policy need for the better collection of oral health status, behavior and use of oral health services in populations studies which are linked into general health and welfare national data reporting.

The data reported in the three Tables however are well presented and very useful for readers.

Specific issues which the authors could/should address would include:

  1. Title: Query change title "social determinants" to "socio-economic determinants" as many of the drivers of cultural and ethnic variation do not appear to be included within the EPICES score.
  2. Abstract: "more likely to consult a dentist" is not consistent with the term used elsewhere "use of dental services."
  3. Introduction: Perhaps citations that include a more complete range of social and behavioral variables influencing access to dental care, including pain, appearance, perceived value of teeth etc  could be referenced in this section.
  4. Introduction: A comment on how the French system determines eligibility for the "Public supplemental insurance" would be valuable for readers.
  5. Introduction: Perhaps clarification of the model driving the analysis and research questions could be addressed in the paragraph before the Aims (ie a cut and paste from paragraph 1 under 2.2).
  6. Materials and Methods: Each of he acronyms needs to be clearly spelt put - IRDES (and later) EPICES. 
  7. Materials and Methods: Perhaps a very brief comment on the value of the EPICES system over say social support tools such as the Duke Social Support Index may be useful here?
  8. Tables: Slightly different wording is used in different Tables to categorize the EPICES Score - please check and make consistent.
  9. Materials and Methods:  The final paragraph (lines 147 - 157) is somewhat confusing - could this be clarified ?
  10. Results: Not sure what "on a contrary ..."means? (lines 164 - 166).
  11. Discussion: Paragraph 3.  Given the weakness you have already identified, I think a more cautious approach should be made to statements such as "Our second most important finding..." there are many variables both behavioral, cultural and ethnic which the research has not included, therefore the language used to describe the relevance of your findings should be approached in a more cautious fashion.
  12. Conclusions: The two issues here are: firstly, more caution, and secondly, identifying where the next step in this research should go.  To me the opening sentence of the Conclusions may better read: "In this study, socio-economic disadvantage, appeared to be the greater barrier to seeking care than to engaging in on-going treatment".  I would delete the next sentence but add in its place: "Public supplemental health insurance was a significant factor in improving access to dental care." I would also suggest the sentence related to the Australian system is better placed in the Discussion than the Conclusions.
  13. Informed Consent Statement:  It is not clear whether the data-bases used were available to the public or whether they required approval to be accessed.  If dat required access approval then I believe that it is mandatory to include a statement which confirms that the use of NIH and IRDES data has been as authorized by the appropriate agency.  Please check this with the Journal Editorial staff.

Author Response

Dear Reviewer,

Thank you so much for your valuable comments.

We applied all your advice in the new document.

We ask for a professional English language check and correct the discrepancies you marked.

We hope this new document is going to be suitable for publication.

Kind Regards.

Round 2

Reviewer 3 Report

Thank you for giving me this opportunity to re-review your revised manuscript.

I am happy that all of the suggested corrections have been made.

Thank you for spending so much time for revised manuscript.

Reviewer 4 Report

The authors have complied with all of the suggestions for change which this Referee recommended. I am satisfied with the changes and have no further advice.